# The Roles of ROS Generation in RANKL-Induced Osteoclastogenesis: Suppressive Effects of Febuxostat

**DOI:** 10.3390/cancers12040929

**Published:** 2020-04-09

**Authors:** Mohannad Ashtar, Hirofumi Tenshin, Jumpei Teramachi, Ariunzaya Bat-Erdene, Masahiro Hiasa, Asuka Oda, Kotaro Tanimoto, So Shimizu, Yoshiki Higa, Takeshi Harada, Masahiro Oura, Kimiko Sogabe, Shingen Nakamura, Shiro Fujii, Ryohei Sumitani, Hirokazu Miki, Kengo Udaka, Mamiko Takahashi, Kumiko Kagawa, Itsuro Endo, Eiji Tanaka, Toshio Matsumoto, Masahiro Abe

**Affiliations:** 1Department of Orthodontics and Dentofacial Orthopedics, Tokushima University Graduate School of Oral Sciences, Tokushima 770-8503, Japan; c301351010@tokushima-u.ac.jp (M.A.); c301751004@tokushima-u.ac.jp (K.T.); c301851012@tokushima-u.ac.jp (S.S.); c301951005@tokushima-u.ac.jp (Y.H.); 2Department of Hematology, Endocrinology and Metabolism, Institute of Biomedical Sciences, Tokushima University, Tokushima 770-8503, Japan; oda.asuka@tokushima-u.ac.jp (A.O.); takeshi_harada@tokushima-u.ac.jp (T.H.); m.oura@tokushima-u.ac.jp (M.O.); sogabe.kimiko@tokushima-u.ac.jp (K.S.); shingen@tokushima-u.ac.jp (S.N.); fujii.shiro@tokushima-u.ac.jp (S.F.); ryoheisumitani@tokushima-u.ac.jp (R.S.); kengo.udaka@gmail.com (K.U.); m.takahashi@tokushima-u.ac.jp (M.T.); kkag@tokushima-u.ac.jp (K.K.); 3Department of Orthodontics and Dentofacial Orthopedics, Institute of Biomedical Sciences, Tokushima University, Tokushima 770-8503, Japan; mhiasa@tokushima-u.ac.jp (M.H.); etanaka@tokushima-u.ac.jp (E.T.); 4Department of Tissue Regeneration, Institute of Biomedical Sciences, Tokushima University, Tokushima 770-8503, Japan; jumptera@tokushima-u.ac.jp; 5Department of Immunology, School of Bio-Medicine, Mongolian National University of Medical Sciences, Ulaanbaatar 14210, Mongolia; ariunzaya.b@mnums.edu.mn; 6Division of Transfusion Medicine and Cell Therapy, Tokushima University Hospital, Tokushima 770-8503, Japan; mikihiro@tokushima-u.ac.jp; 7Department of Chronomedicine, Institute of Biomedical Sciences, Tokushima University, Tokushima 770-8503, Japan; endoits@tokushima-u.ac.jp; 8Fujii Memorial Institute of Medical Sciences, Tokushima University, Tokushima 770-8503, Japan; toshio.matsumoto@tokushima-u.ac.jp

**Keywords:** multiple myeloma, osteoclastogenesis, RANKL, ROS, doxorubicin, ovariectomy

## Abstract

Receptor activator of NF-κB ligand (RANKL), a critical mediator of osteoclastogenesis, is upregulated in multiple myeloma (MM). The xanthine oxidase inhibitor febuxostat, clinically used for prevention of tumor lysis syndrome, has been demonstrated to effectively inhibit not only the generation of uric acid but also the formation of reactive oxygen species (ROS). ROS has been demonstrated to mediate RANKL-mediated osteoclastogenesis. In the present study, we therefore explored the role of cancer-treatment-induced ROS in RANKL-mediated osteoclastogenesis and the suppressive effects of febuxostat on ROS generation and osteoclastogenesis. RANKL dose-dependently induced ROS production in RAW264.7 preosteoclastic cells; however, febuxostat inhibited the RANKL-induced ROS production and osteoclast (OC) formation. Interestingly, doxorubicin (Dox) further enhanced RANKL-induced osteoclastogenesis through upregulation of ROS production, which was mostly abolished by addition of febuxostat. Febuxostat also inhibited osteoclastogenesis enhanced in cocultures of bone marrow cells with MM cells. Importantly, febuxostat rather suppressed MM cell viability and did not compromise Dox’s anti-MM activity. In addition, febuxostat was able to alleviate pathological osteoclastic activity and bone loss in ovariectomized mice. Collectively, these results suggest that excessive ROS production by aberrant RANKL overexpression and/or anticancer treatment disadvantageously impacts bone, and that febuxostat can prevent the ROS-mediated osteoclastic bone damage.

## 1. Introduction

Multiple myeloma (MM) has a unique propensity to almost exclusively develop in the bone marrow and generates devastating bone destruction. MM cells enhance osteoclast (OC) formation and activity and suppress osteoblastic differentiation from bone marrow stromal cells, leading to extensive bone destruction with rapid loss of bone [1,2,3]. Receptor activator of NF-κB ligand (RANKL), a critical mediator of osteoclastogenesis, is upregulated in bone marrow stromal cells to extensively enhance osteoclastogenesis and bone resorption in MM; importantly, activated OCs in turn enhance glycolysis in MM cells and thereby MM cell proliferation, leading to the formation of a progressive vicious cycle between MM tumor expansion and osteoclastic bone destruction [4,5,6]. Therefore, OCs should be targeted to improve treatment efficacy, especially in MM expanding in the bone marrow with enhanced bone resorption. 

Reactive oxygen species (ROS) is induced and plays important roles in a variety of pathological cellular processes [7,8,9]. ROS has been demonstrated to be produced during RANKL-induced osteoclastogenesis from bone marrow monocyte–macrophage lineage cells (BMMs), and antioxidants, including N-acetylcysteine (NAC), have been proven to prevent the RANKL-induced OC differentiation by decreasing ROS [10,11,12]. These results indicate that ROS plays a critical role in OC differentiation and activity [12,13,14]. In cancer treatment, ROS can be generated under stressful conditions by anticancer agents. Anticancer agents, including doxorubicin (Dox) and proteasome inhibitors, induce excessive levels of ROS in cancer cells to cause cell death [15,16]. Besides in cancer cells, these anticancer agents generate ROS in different types of normal cells, including BMM, and such generated ROS might facilitate RANKL-mediated osteoclastogenesis in bone-residing cancers with RANKL upregulation in the bone marrow, including MM and cancer metastasis to bone [17]. However, the roles of ROS induced by anticancer agents in osteoclastogenesis have not been studied.

Febuxostat, a selective and potent inhibitor of xanthine oxidase (XO), has been approved and clinically used for prevention of tumor lysis syndrome, a life-threatening oncologic emergency, in patients with malignant tumors receiving chemotherapy [18]. Febuxostat has been demonstrated to effectively inhibit not only the generation of uric acid but also the formation of ROS [19,20]. In the present study, we therefore explored the role of cancer-treatment-induced ROS in RANKL-mediated osteoclastogenesis and the suppressive effects of febuxostat on ROS generation and osteoclastogenesis. Here, we demonstrated that Dox and RANKL cooperatively enhance osteoclastogenesis through ROS production, and that febuxostat effectively suppresses osteoclastogenesis enhanced by Dox and RANKL in combination as well as in cocultures with MM cells. These results suggest that excessive ROS production by aberrant RANKL overexpression in MM and/or anticancer treatment disadvantageously impacts bone, leading to pathological bone damage and cancer-treatment-induced bone loss (CTIBL). The therapeutic impact of febuxostat can be expected against cancer-induced pathological bone damage and CTIBL.

## 2. Results

### 2.1. Febuxostat Inhibits RANKL-Induced ROS Production and OC Formation

As previously reported [11,21], RANKL dose-dependently induced ROS production in RAW264.7 preosteoclastic cells (Figure 1A). Since the XO inhibitor febuxostat has been demonstrated to inhibit XO-mediated intracellular ROS production [19], we looked at the effects of febuxostat on ROS production and OC formation. To confirm the effects of febuxostat on ROS production at cellular levels, we counted ROS-expressing cell numbers using H2DCFDA staining. Treatment with RANKL increased the numbers of ROS-expressing RAW264.7 cells; however, Figure 1B shows that febuxostat at 10 μM was able to mostly inhibit the appearance of the ROS-expressing cells. We further examined the effects of febuxostat at different concentrations. Febuxostat dose-dependently suppressed ROS production and almost completely at 60 μM (Figure 1C). Febuxostat at 60 μM did not affect the viability of RAW264.7 cells as well as primary bone marrow cells (Figure 1D). Therefore, we thereafter set the concentration of febuxostat to be 60 μM for the in vitro experiments. Febuxostat at 60 μM suppressed RANKL-induced expression of OC differentiation markers: c-Fos, cathepsin K (CTSK), and NFATc1 (Figure 1E). Moreover, OC formation and activity (resorption areas) were also potently suppressed with febuxostat at 60 μM (Figure 1F). Because ROS has been recognized as an intracellular signal mediator for RANKL-induced OC differentiation [11,21,22], these results suggest that febuxostat can act as an inhibitor for RANKL-induced osteoclastogenesis through the suppression of ROS production.

### 2.2. Dox Facilitates RANKL-Mediated Osteoclastogenesis Through ROS Production

Induction of ROS is among the predominant cytotoxic mechanisms of anticancer agents [23,24]. Dox is an important chemotherapeutic agent in treatment against lymphoid malignancies, including MM [25]. However, the induction of ROS in microenvironmental cells surrounding cancer cells and the effects of the induced ROS on their cellular function have not been precisely studied. Because RANKL expression is upregulated to extensively enhance osteoclastic bone destruction in MM [5,6], we next explored the effects of Dox on ROS production in osteoclastic lineage cells and thereby osteoclastogenesis upon stimulation with RANKL. Dox alone dose-dependently induced ROS production in RAW264.7 cells, which was suppressed by the addition of febuxostat (Figure 2A). Dox further upregulated their RANKL-induced ROS production (Figure 2B), suggesting cooperative generation of ROS by Dox and RANKL in combination. However, febuxostat was able to effectively suppress the ROS production by Dox and RANKL in combination. Interestingly, Dox and RANKL cooperatively induced NFATc1 expression in RAW264.7 cells, which was also suppressed by febuxostat (Figure 2C). Besides febuxostat, NAC, an ROS scavenger, similarly reduced ROS production and NFATc1 induction in RAW264.7 cells upon treatment with Dox or RANKL in combination (Figure 2D), further indicating the critical roles of ROS production. Intriguingly, febuxostat as well as NAC induced NFATc1 expression in the absence of Dox and RANKL. However, *NFATC1* mRNA expression levels were rather suppressed with febuxostat (Appendix A). Redox status under febuxostat or NAC may affect stabilization of NFATc1 protein, which should be further studied. Importantly, Dox and RANKL cooperatively enhanced in vitro osteoclastogenesis from primary bone marrow cells and their bone resorptive activity, which was abolished by the addition of febuxostat (Figure 2E). However, addition of Dox did not enhance bone resorptive activity of re-plating osteoclasts at per cell levels in the presence of RANKL, while febuxostat was able to suppress the bone resorbing activity of osteoclasts (Appendix A). Therefore, the enhancement of bone resorptive activity by Dox (Figure 2E) appears to be due to an increase in numbers of differentiated osteoclasts. In addition, treatment with febuxostat either for days 1 and 2 or for days 5–10 was able to suppress osteoclast formation by RANKL alone (Appendix A). Treatment with Dox from days 5–10 enhanced osteoclast formation by RANKL, whereas the treatment for the first 2 days did not affect it (Appendix A). Febuxostat also suppressed the Dox’s enhancement of osteoclast formation. Precise mechanisms of induction of osteoclastogenesis by Dox in the presence of RANKL remain to be clarified. These results suggest that further accumulation of ROS by Dox facilitates RANKL-mediated osteoclastogenesis and that febuxostat can effectively suppress the ROS production and thereby osteoclastogenesis induced by Dox and RANKL in combination.

### 2.3. Febuxostat Does Not Compromise the Cytotoxic Effects of Dox on MM Cells

ROS induction is regarded as an important mechanism of anticancer effects by Dox [26]. Treatment with Dox induced ROS production from MM cells (Appendix A). Given the suppression of ROS induction by febuxostat, febuxostat may mitigate the anticancer effects of Dox. We therefore asked whether or not febuxostat affects the cytotoxic effects of Dox against MM cells. Intriguingly, febuxostat only partially but dose-dependently impaired the viability of MM cells (Figure 3A). Dox dose-dependently reduced the viability of MM cells, and the Dox’s cytotoxic effects mostly remained in the presence of febuxostat, although Dox and febuxostat at 50 μM did not cooperatively induce cell death in RPMI8226, U266, OPM-2, and MM.1S cells. In addition, Dox induced the activation of caspase 3 in MM cells, which was not compromised by febuxostat (Figure 3B). Because Dox facilitated osteoclastogenesis without apparently inducing cell death in combination with RANKL, we next examined the cytotoxic effects of Dox on mature OCs in comparison with those on MM cells. Primary bone marrow cells were cultured with RANKL for 10 days to give rise to mature OCs as determined by TRAP-positive large multinucleated cells or by F-actin ring formation. Addition of Dox for 2 days did not reduce the numbers of TRAP-positive multinucleated cells as well as cells with F-actin ring formation on dishes (Appendix A). We previously reported that PIM2 is an important regulator of RANKL-mediated OC function and activity [27,28]. Mature OCs were tightly adhered to plastic dishes and remained after pipetting and washing. The mature OCs were harvested after pipetting and washing, and their expression of PIM2 was investigated under the Dox treatment. Dox dose-dependently induced the expression of PIM2 in mature OCs as RANKL did (Appendix A). Although febuxostat dose-dependently induced MM cell death and did not impair the Dox’s cytotoxic activity, the ROS scavenger NAC alone did not induce MM cell death and appeared to compromise Dox’s cytotoxic activity at the concentration of 2.5 mM (Figure 3C). These effects of febuxostat seem unique, which should be further studied. These results suggest that osteoclastic lineage cells appear to be less sensitive to Dox compared with MM cells, and that febuxostat suppresses MM cell viability without compromising Dox’s anti-MM activity. 

### 2.4. Febuxostat Suppresses Osteoclastogenesis by MM Cells

MM exhibits potent osteoclastic bone destruction through upregulation of RANKL expression in bone marrow stromal cells in the bone marrow [5,6]. We next examined the effects of febuxostat on in vitro osteoclastogenesis in cocultures of MM cells with primary bone marrow cells. The human MM cell lines RPMI8226 and KMS-11 and the murine MM cell line 5TGM1 enhanced the formation of TRAP-positive multinucleated OCs, when cocultured with whole bone marrow cells (Figure 4A). However, treatment with febuxostat effectively suppressed the osteoclastogenesis in the cocultures with the MM cells. Because MM cells upregulate RANKL expression in the bone marrow stroma cells to enhance osteoclastogenesis [29], we next asked whether febuxostat affects induction of RANKL expression in bone marrow stromal cells in the cocultures with MM cells. After coculturing MM cells, nonadherent cells were washed and removed. Remaining adherent bone marrow stromal cells were collected and the expression of the *RANKL* gene was examined. The cocultures of MM cells induced *RANKL* expression in bone marrow stromal cells; however, febuxostat abolished the *RANKL*, indicating the suppression of *RANKL* upregulation by febuxostat (Figure 4B). Doxorubicin alone did not induce the expression of *RANKL* mRNA in bone marrow stromal cells (Figure 4C). These results suggest that febuxostat may directly act on osteoclastic lineage cells and also impair RANKL expression in bone marrow stromal cells to suppress osteoclastogenesis in cocultures of MM cells with bone marrow cells.

### 2.5. Febuxostat Reduces Osteoclastic Activity and Bone Loss in Ovariectomized (OVX) Mice

The impact of ovariectomy on the bone antioxidant system has been demonstrated [30,31,32]. Therefore, we next explored the therapeutic potential of febuxostat in OVX-induced osteoporosis models. Febuxostat was administered every day for 3 weeks in OVX or sham-operated mice. μCT images of the tibiae revealed the reduction of trabecular bone volume in OVX mice compared with that in sham-operated mice (Figure 5A). However, the trabecular bone structure and volume were not decreased in OVX mice treated with febuxostat. Histomorphometric analyses on the tibiae showed that the values of bone volume/total volume (BV/TV) and trabecular number (Tb.N) were decreased while those of trabecular separation (Tb.Sp) were increased in OVX mice (Figure 5B). However, treatment with febuxostat significantly ameliorated these OVX-induced changes. Moreover, serum levels of the bone resorption marker TRACP-5b were increased (Figure 5C), and the numbers of OCs were increased (Figure 5D) in OVX mice; however, treatment with febuxostat mostly abolished the elevation of the TRACP-5b levels and decreased the numbers of OCs. These results indicate that febuxostat has the therapeutic potential to ameliorate OVX-induced bone loss in vivo. 

## 3. Discussion

RANKL induces ROS generation, which is essential for induction of OC differentiation from OC precursors [10,11,21]. In the present study, we demonstrated that Dox further enhances the accumulation of ROS in osteoclastic lineage cells in combination RANKL, thereby stimulating RANKL-induced osteoclastogenesis. However, febuxostat was shown to effectively inhibit ROS generation in osteoclastic lineage cells by RANKL and suppress the induction of osteoclastogenesis by RANKL. 

Febuxostat is a well-known drug for the reduction of uric acid production and is widely used in patients with hyperuricemia or gout [33]. Recently, febuxostat has been approved for prevention of tumor lysis syndrome in cancer patients receiving chemotherapies [34,35]. Despite its clinical application for the prevention of tumor lysis syndrome in cancer patients, little is known about the effects of febuxostat on cancer growth and pathological conditions related to cancers. We found that febuxostat effectively suppressed the accumulation of ROS in osteoclastic lineage cells by RANKL or Dox alone or both in combination, and that febuxostat treatment is able to prevent RANKL-stimulated in vitro OC formation even in the presence of Dox. Moreover, febuxostat was able to inhibit osteoclastogenesis induced in cocultures of bone marrow cells with MM cells and pathological bone loss in OVX mice.

The majority of cytotoxic agents against cancer cells are generally accepted to kill cancer cells though excessive ROS production [23,36]. These anticancer agents can also induce ROS production in nontumorous bystander cells to cause the development of adverse effects. Based on the present results, excessive ROS production by such anticancer agents may enhance bone resorption, especially in patients with bone cancers and osteoporosis in which RANKL expression is upregulated in their bone marrow. Febuxostat is expected to alleviate ROS production in osteoclastic lineage cells and thereby prevent ROS-induced osteoclastogenesis in patients receiving ROS-inducible anticancer agents such as Dox.

However, reduction of ROS accumulation in cancer cells by antioxidants such as NAC may compromise the cytotoxic activity of anticancer agents, which becomes a major concern for clinical application of antioxidants in cancer treatment. In the present study, we looked at the effects of febuxostat on the viability of MM cells. Intriguingly, febuxostat alone was able to reduce the viability of MM cells and did not apparently antagonize Dox’s cytotoxic activity to allow Dox to exert cytotoxic activity against MM cells (Figure 3A). The underlying mechanisms by which febuxostat reduces tumor cell viability and the combinatory cytotoxic effects of febuxostat and Dox remain to be further clarified. In addition, ROS is produced from different sources such as NADPH oxidase and the mitochondrial electron transport system other than xanthine oxidase. Therefore, we need to clarify the mechanisms of ROS production in MM cells and osteoclasts under different stimuli, and also the effects of agents to suppress different sources of ROS production other than xanthine oxidase.

Cancer patients have a greater chance of living longer owing to recent improvement of anticancer treatment modalities. However, bone loss emerges as one of the most serious unmet issues associated with long-term repeated treatment with anticancer agents, which can be called CTIBL [37]. Bisphosphonate or denosumab are recommended to be administered to prevent cancer-related bone destruction and CTIBL [38,39]. The present study suggests that excessive ROS production by aberrant RANKL overexpression in MM and/or anticancer treatment disadvantageously impacts bones, leading to pathological bone damage and CTIBL. ROS scavenging agents such as febuxostat may help to prevent CTIBL. The preventive activity of febuxostat against CTIBL should be validated in well-designed clinical studies. 

## 4. Materials and Methods

### 4.1. Reagents

The reagents used were bought from the following specified manufacturers: rabbit monoclonal anti-c-fos, cleaved caspase 3, caspase 3, rabbit polyclonal anti-Pim-2, horseradish peroxidase (HRP)-anti-rabbit IgG, and anti-mouse IgG from Cell Signaling Technology (Beverly, MA, USA); mouse monoclonal anti-β-actin antibody from Sigma-Aldrich (St. Louis, MO, USA); mouse monoclonal antibodies against NFATc1 and CTSK from Santa Cruz Biotechnology Inc. (Santa Cruz, CA, USA); human macrophage colony-stimulating factor (M-CSF) from Wako (Osaka, Japan); human soluble RANKL from Oriental Yeast Co. Ltd (Shiga, Japan); febuxostat from TEIJIN (Osaka, Japan); NAC from Nacalai tesqure (Kyoto, Japan); and Dox from Tokyo Chemical Industry (Tokyo, Japan).

### 4.2. Cell Culture

Murine preosteoclastic cell line RAW264.7 cells were cultured in Eagle’s minimal essential medium α modification (Sigma-Aldrich) complemented with 10% FBS, L-glutamine, and 50 mg/mL penicillin/streptomycmo Fisher Scientific, Waltham, MA, USA). The human MM cell lines RPMI8226, KMS-11, MM.1S, U266, and OPM-2 were obtained from the American Type Culture Collection (Rockville, MD, USA). The mouse MM cell line 5TGM1 was a kind gift from Gregory R. Mundy (Vanderbilt Center for Bone Biology, Vanderbilt University, Nashville, TN, USA). MM cells were cultured in RPMI 1640 medium (Sigma-Aldrich) supplemented with 10% FBS, penicillin G at 50 mg/mL, and streptomycin at 50 mg/mL. Human stromal cells were isolated from patient’s samples and cultured in the same culture medium of MM cells. All procedures involving human specimens were performed with written informed consent according to the Declaration of Helsinki and using a protocol approved by the Institutional Review Board for human protection at Tokushima University Hospital (240-2).

### 4.3. OC Formation

OCs were produced from murine preosteoclastic cell line RAW264.7 cells or mouse bone marrow cells as previously explained [40,41]. Whole bone marrow cells were harvested from the femur of C57BL/6J mice (SLC, Tokyo, Japan), and nonadherent cells were collected and cultured with M-CSF (10 ng/mL) for 3 days to generate primary BMMs. Then, the BMMs were cultured for 10–14 days with M-CSF (10 ng/mL) and RANKL (50 ng/mL) to generate mature OCs. Culture media was changed every 2 days. To investigate the effects of febuxostat or Dox on osteoclastogenesis and bone resorption, BMMs were cultured with M-CSF (10 ng/mL) and RANKL (50 ng/mL) in the presence or absence of febuxostat or Dox for 10–14 days. TRAP-positive cells were detected with a Leukocyte Acid Phosphatase Assay kit (Sigma-Aldrich). TRAP-positive cells containing 3 or more nuclei were counted as OCs under a light microscope (BX50; Olympus, Tokyo, Japan). 

### 4.4. Bone Resorption Assay

The effect of febuxostat or Dox on RANKL-induced bone resorption was analyzed using Corning Osteo-Assay Surface 96-well plates (Corning, Lowell, MA, USA), as we described previously [27]. BMMs were cultured in α-MEM containing 10% FBS with M-CSF (10 ng/mL) and RANKL (25 or 50 ng/mL) in the presence or absence of febuxostat (60 μM) or Dox (0.1 μM). After 14 days of culture, the attached cells were removed from the slides using 10% sodium hypochlorite. The resorbed area of bone resorption was determined using image analysis techniques (National Institutes of Health ImageJ system; http://imagej.nih.gov/ij/).

### 4.5. Coculture Experiment

Whole bone marrow cells harvested from C57BL/6J mice were seeded onto a 24-well plate and cultured with M-CSF (10 ng/mL) to generate osteoclast precursors for 3 days. Then, human MM cell lines (RPMI8226 and KMS-11) and the murine MM cell line 5TGM1 were added at 1 × 10^4^ cells per well. The cells were cultured with or without febuxostat at 60 μM for 14 days in the presence of M-CSF (10 ng/mL). The numbers of TRAP-positive multinucleated cells per well were counted under a light microscope (BX50; Olympus, Tokyo, Japan).

### 4.6. Animal Experiment

Experiments on animals were performed under the regulation and permission of the Animal Care and Use Committee of Tokushima University, Tokushima, Japan (T30-3). Sixteen female 8-week-old C57BL/6J mice were purchased from SLC (Tokyo, Japan). Sham operation (*n* = 8) and bilateral ovariectomy (*n* = 8) were performed following previous studies [42,43]. Then, the mice were randomly divided into four groups: sham mice (*n* = 4), OVX mice (*n* = 4), sham mice with febuxostat administration (*n* = 4), and OVX mice with febuxostat administration (*n* = 4). Febuxostat was administered orally at 5 mg/kg every day for 3 weeks. The mice were then sacrificed and bone and serum were preserved for analysis. Micro-CT (μCT) images of tibiae were taken (SkyScan 1176 scanner and associated analysis software; Buruker, Billerica, Massachusetts, USA). The trabecular bone analysis was performed on μCT images to calculate the following bone morphometric parameters: bone volume density (BV/TV), trabecular number (Tb.N), and trabecular separation (Tb.Sp). The expression of TRACp-5b was measured to analyze OC activity by obtaining blood serum samples with the TRACP & ALP Assay Kit (Takara Bio., Siga, Japan).

### 4.7. Quantifying ROS

To detect the ROS expression in the cells, CellROX™ Green Reagent or H2DCFDA reagent (Life Technologies, Carlsbad, CA, USA) were used and the manufacturer’s instructions were followed. Briefly, RAW264.7 cells (2 × 10^4^ cells/well) were seeded in a glass-bottom 96-well plate, and the cells were pretreated with febuxostat or NAC for 2 h. Then, ROS-inducing reagents RANKL (final concentration was 25 or 50 ng/mL) and/or doxorubicin (final concentration was 0.1–1.0 μM) were added. Then, CellRox Green (Invitrogen, Carlsbad, CA, USA) was added in a final concentration of 5 μM and returned to the incubator for 30 min. Afterwards, the wells were washed using PBS and the plate was placed in the microplate reader SpectraMax i3 (Molecular Devices, San Jose, CA, USA) to obtain the results. The results were expressed as fold changes from control. To detect the ROS with the H2DCFDA reagent, the cells were pretreated with H2DCFDA reagent at 10 μM and 37 °C, followed by RANKL (50 ng/mL) stimulation. The intracellular ROS was detected as DCF with fluorescence. The DCF-positive cells were counted as ROS-expressing cells under a fluorescence microscope (BZ-X800; Keyence, Osaka, Japan).

### 4.8. Reverse Transcription Polymerase Chain Reaction (RT-PCR) and Real-time PCR

RNA was obtained using TRIZOL reagent (Gibco BRL, Maryland, USA). Two micrograms of total RNA were reverse-transcribed with Superscript II (Gibco) in 20 µL of reaction solution. One-tenth of the RT-PCR products was used for subsequent PCR analysis with 23–35 cycles at 95 °C for 30 s, 58 °C for 30 s, and 72 °C for 30 s. Gene expression levels were detected using ethidium bromide as a fluorescent dye. 

For real-time RT-PCR, each cDNA sample was amplified using SYBR Premix EX Taq II (Takara Bio Inc., Shiga, Japan) on the 7300 Real-time PCR System (Thermo Fisher Scientific). Briefly, the reaction conditions consisted of 2 μL of cDNA and 0.4 μM primers in a total volume of 20 μL. *GAPDH* was used as an endogenous control to normalize each sample.

The following primer sequences were used: h*GAPDH* F: TGTCTTCACCACCATGGAGAAGG, R: GTGGATGCAGGGATGATGTTCTG and h*RANKL* F: GGATCACAGCACATCAGAGCAGAG, R: GTAAGGAGGGGTTGGAGACCTCG.

### 4.9. Cell Viability

Cell viability was determined using the Cell Counting Kit-8 assay (Dojindo, Kumamoto, Japan) according to the manufacturer’s instructions. MM cells were cultured on a 96-well plate with febuxostat or Dox for 48 h and then incubated with 2-(2-methoxy-4- nitrophenyl)-3-(4-nitrophenyl)-5-(2,4-disulphophenyl)-2H-tetrazolium monosodium salt (WST-8) for 3 h. Then, the absorbance of each well was measured at 450–655 nm with an iMark microplate reader (Bio-Rad Laboratories, Hercules, CA, USA).

### 4.10. Western Blot

Radioimmunoprecipitation assay RIPA buffer was used to isolate cell extracts; 1.0 mL was taken from the buffer and supplemented with protease inhibitor (10 μL), phosphatase inhibitor (10 μL), dithiothreitol (DTT) (1 μL), and phenylmethylsulfonyl fluoride (PMSF) (10 μL). The isolated protein concentration from each sample was measured using the Pierce™ BCA Protein Assay kit (Bio-Rad, Hercules, CA, USA) and the concentration was adjusted. Then, the samples were heated to 95 °C for 5 min. The cell lysates and size marker (Wide-View Prestained Protein Size Marker III, Wako, Osaka, Japan) were separated by sodium dodecyl sulfate polyacrylamide gel electrophoresis on a 10% polyacrylamide gel and transferred to polyvinylidene difluoride membranes (Millpore, Billerica, MA, USA). The membranes were then blocked in 3% skimmed milk and placed in primary antibodies overnight. Then, a secondary antibody appropriate for the primary was used for each membrane, after which the membrane was examined using Amersham Imager 600 (GE Life Sciences, Little Chalfont, England) in fluorescence mode and the data were analyzed. The band intensities from each blot were quantified using ImageJ software (NIH) and normalized to β-actin, and then the fold change from the control was calculated and shown.

### 4.11. Actin Ring Staining

BMMs were cultured with M-CSF (10 ng/mL) and RANKL (50 ng/mL) to generate mature OCs. Mature OCs were treated with Dox (0.1, 0.5, 1.0 μM) for 48 h. Then, the OCs were stained with Acti-stain^TM^ 488 phalloidin (Cytoskeleton, Inc., Denver, USA) following the manufacturer’s procedure. The number of OCs with F-actin ring per well was counted under a fluorescence microscope (BZ-X800; Keyence, Osaka, Japan)

### 4.12. Immunofluorescence Staining

To calculate the number of OCs in OVX mice, the tissue sections (5 μm) of tibia were incubated with anti-CTSK antibody (1:100) overnight at 4 °C after deparaffinization and blocking (3% bovine serum, 1 h). Tissue sections were washed in PBS and incubated with Alexa Fluor^®^ 488-conjugated anti-mouse IgG secondary antibodies (1:200 in PBS; Life technologies, California, USA) for 1 h. The wide field-of-view fluorescence images were examined using a fluorescence microscope (BZ-X800; Keyence, Osaka, Japan) and the number of CTSK-positive OCs per field was counted. Original magnification was ×100.

### 4.13. Statistical Analysis

We completed statistical analysis using Student’s *t* test or ANOVA, where *p* < 0.05 was considered as a significant difference. In this study, all statistical analyses were performed using Statcel 4 Software (OMS Publishing, Saitama, Japan). 

## 5. Conclusions 

The anticancer agent Dox further accumulated ROS to facilitate RANKL-mediated osteoclastogenesis. Febuxostat effectively suppressed the ROS production and thereby osteoclastogenesis by Dox and RANKL in combination. Importantly, febuxostat was able to inhibit osteoclastogenesis enhanced in cocultures of bone marrow cells with MM cells and alleviated pathological bone loss in ovariectomized mice. In addition, febuxostat rather suppressed MM cell viability without compromising Dox’s anti-MM activity. Therefore, a therapeutic impact of febuxostat can be expected against cancer-induced pathological bone damage and CTIBL.

## Figures and Tables

**Figure 1 cancers-12-00929-f001:**
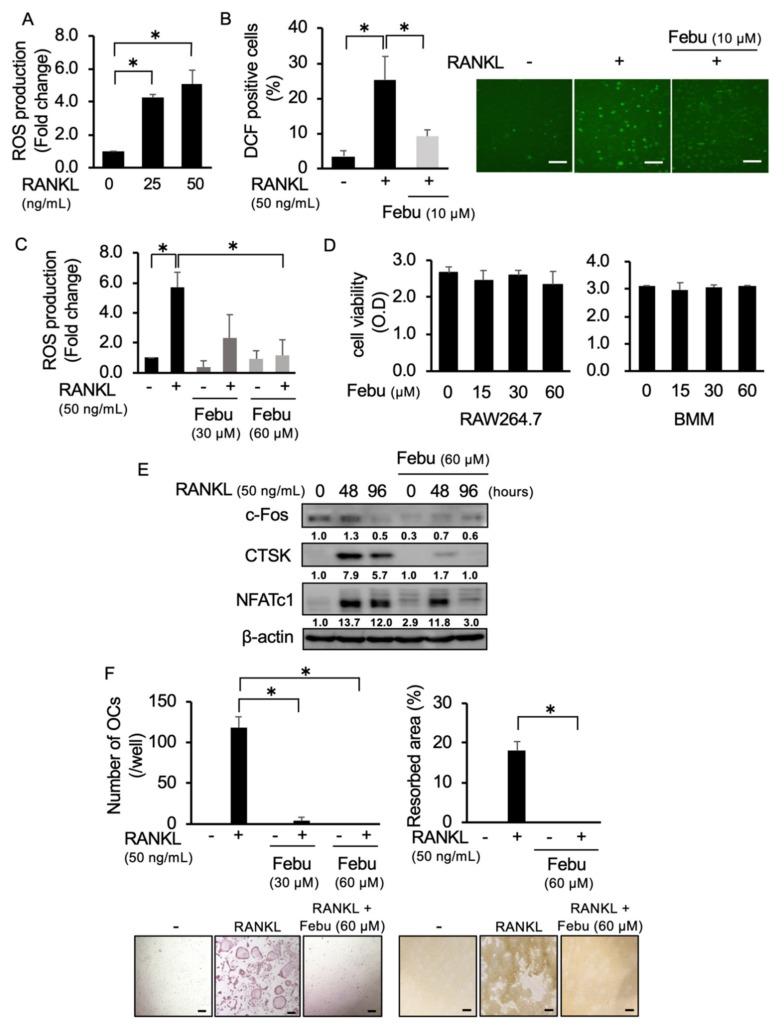
Febuxostat (Febu) inhibits receptor activator of NF-κB ligand (RANKL)-induced reactive oxygen species (ROS) production and osteoclast (OC) formation. (**A**) RAW264.7 cells were seeded onto a glass-bottom 96-well plate and cultured in quadruplicate with indicated concentrations of RANKL for 30 min followed by staining with the ROS detector CellRox green. Data are expressed as fold changes from controls (mean ± SD). (**B**) RAW264.7 cells were seeded onto a glass-bottom 96-well plate and cultured with RANKL (50 ng/mL) with or without Febu at 10 μM for 30 min. The number of RAW264.7 cells per field was counted under bright-field using a fluorescence microscope (BZ-X800) and ROS-expressing cells were defined as fluorescence DCF-positive cells (green). Data are expressed as % distribution of ROS-expressing cells (per field), mean ± SD. Representative photos are shown. Original magnification, ×200. Bar, 100 μm. (**C**) RAW264.7 cells were seeded onto a glass-bottom 96-well plate and cultured in triplicate with RANKL (50 ng/mL) with or without the indicated concentrations of Febu for 30 min, followed by CellRox green staining. Data are expressed as fold changes from controls (mean ± SD). (**D**) RAW264.7 cells and bone marrow monocyte–macrophage lineage cells (BMMs) were cultured in quadruplicate for 24 h and 10 days, respectively. Febu was added at the indicated concentrations. Cell viability was measured using the 2-(2-methoxy-4- nitrophenyl)-3-(4-nitrophenyl)-5-(2,4-disulphophenyl)-2H-tetrazolium monosodium salt (WST-8) assay. Data are expressed as mean ± SD. (**E**) BMMs were cultured with macrophage colony-stimulating factor (M-CSF) (10 ng/mL) and RANKL (50 ng/mL) in the presence or absence of Febu (60 μM). Cell lysates were collected at 48 and 96 h. c-Fos, cathepsin K (CTSK), and NFATc1 protein levels were analyzed by Western blotting. β-actin served as a loading control. The band sizes of NFATc1 were densitometrically compared to those of a control after normalization to those of β-actin. (**F**) BMMs were cultured in quadruplicate with M-CSF (10 ng/mL) and RANKL (50 ng/mL) in the presence or absence of indicated concentrations of Febu for 10 days. TRAP-positive cells containing three or more nuclei per well were counted (left). Bone resorption activity was also analyzed and results are expressed as % resorbed area (right). Data are presented as mean ± SD. * *p* < 0.05. Representative photos are shown. Original magnification, ×100. Bar, 100 μm.

**Figure 2 cancers-12-00929-f002:**
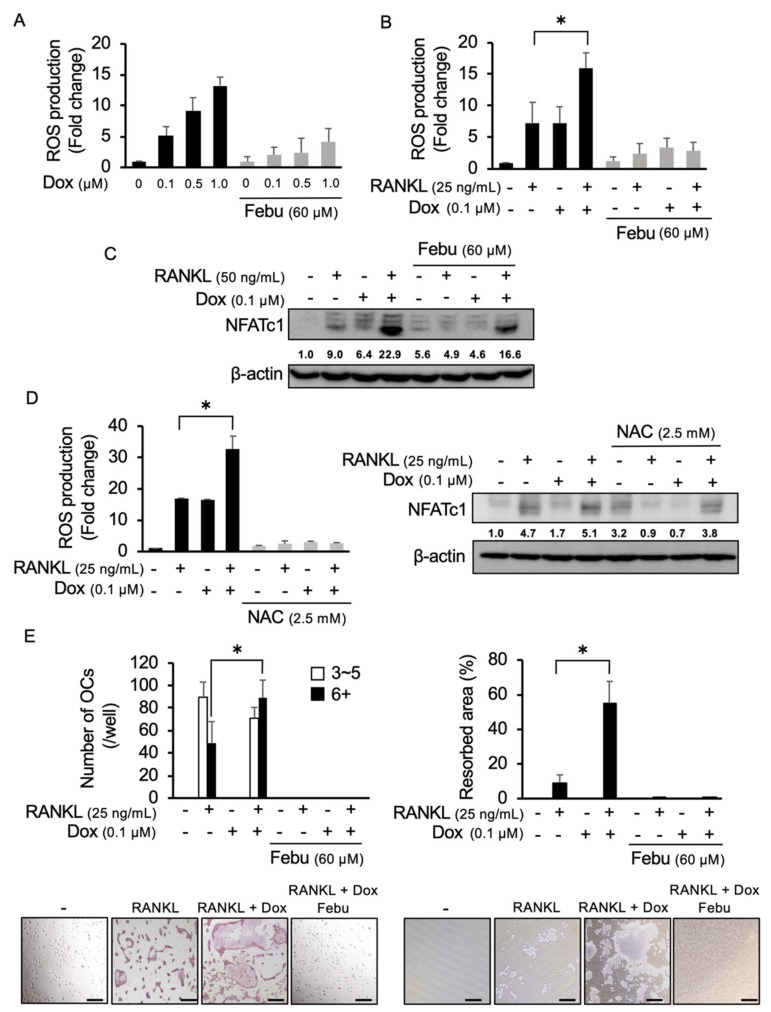
ROS production and osteoclastogenesis by Dox and RANKL in combination. (**A**) RAW264.7 cells were cultured in quadruplicate with indicated dose of doxorubicin (Dox) in the presence or absence of febuxostat (Febu) at 60 μM for 30 min. ROS expression was detected by CellRox green staining. Data are expressed as fold changes from controls (mean ± SD). (**B**) RAW264.7 cells were cultured in quadruplicate with Dox and/or RANKL as indicated for 30 min, and ROS expression was detected by CellRox green staining. Data are expressed as fold changes from controls (mean ± SD). (**C**) RAW264.7 cells were cultured with indicated reagents for 48 h. NFATc1 levels were analyzed by Western blotting. β-actin served as a loading control. The band sizes of NFATc1 were densitometrically compared to those of a control after normalization to those of β-actin. (**D**) RAW264.7 cells were cultured in quadruplicate with indicated reagents for 30 min and ROS expression was detected by CellRox green staining (left). Data are expressed as fold changes from controls (mean ± SD). * *p* <0.05. RAW264.7 cells were cultured with indicated reagents for 48 h. NFATc1 protein levels were analyzed by Western blotting (right). β-actin served as a loading control. (**E**) Bone marrow macrophages (BMMs) were cultured in quadruplicate with M-CSF (10 ng/mL) and indicated reagents for 14 days. The numbers of TRAP-positive multinucleated cells with 3-5 nuclei (white bars) or 6 or more nuclei (black bars) per well were counted. Bone resorption activity was analyzed and the results are presented as % resorbed area. Data are expressed as mean ± SD. * *p* < 0.05. Representative photos are shown. Original magnification, ×100. Bar, 100 μm.

**Figure 3 cancers-12-00929-f003:**
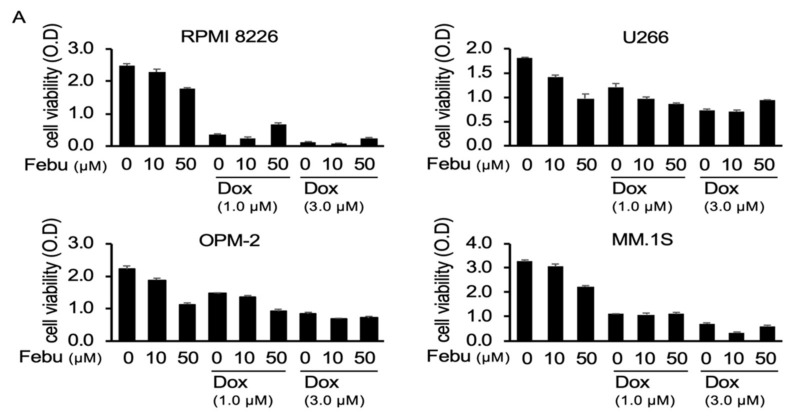
Febuxostat does not compromise the cytotoxic effects of Dox on multiple myeloma (MM) cells. (**A**) The MM cell lines RPMI82266, U266, OPM-2, and MM.1S were cultured in triplicate with the indicated concentrations of Dox and febuxostat (Febu) for 48 h. The cell viability was measured by the WST-8 cell proliferation assay. Results are expressed as mean ± SD. (**B**) RPMI82266 and MM.1S MM cells were cultured with the indicated concentrations of Dox and Febu for 24 h. Caspase 3 and cleaved caspase 3 protein levels were analyzed by Western blotting. β-actin served as a loading control. (**C**) RPMI8226 and U266 MM cells were cultured in triplicate with the indicated concentrations of Dox and/or NAC for 48 h. The cell viability was analyzed by the WST-8 cell proliferation assay. Results are expressed as mean ± SD.

**Figure 4 cancers-12-00929-f004:**
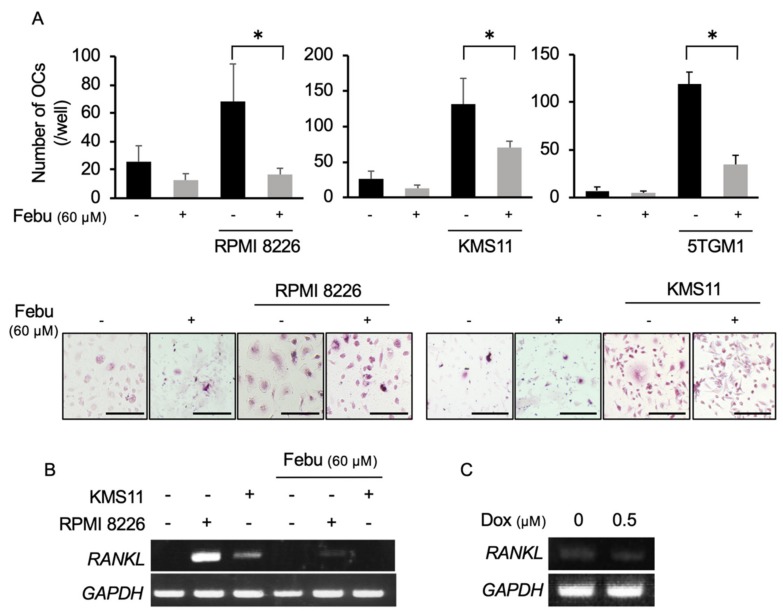
Febuxostat suppresses osteoclastogenesis by MM cells. (**A**) Mouse whole bone marrow cells were seeded onto 24-well plates and cultured with M-CSF (10 ng/mL) for 3 days. After that, human MM cell lines (RPMI8226 and KMS-11) and the murine MM cell line 5TGM1 were added onto the cells. The cells were cultured in quadruplicate with or without febuxostat (Febu) (60 μM) for 14 days in the presence of M-CSF (10 ng/mL). The numbers of TRAP-positive multinucleated cells per well were counted. Data are expressed as mean ± SD. * *p* < 0.05. Representative photos are shown. Original magnification, ×200. Bar, 100 μm. (**B**) Human bone marrow stromal cells were cultured with RPMI 8226 and KMS-11 MM cell lines in the presence or absence of Febu at 60 μM for 24 h. After removing the MM cells, the expression of *RANKL* in the bone marrow stromal cells was detected by RT-PCR. *GAPDH* served as an internal control. (**C**) Human bone marrow stromal cells were cultured with Dox at 0.5 μM for 24 h. The expression of *RANKL* mRNA was detected by RT-PCR. *GAPDH* served as an internal control.

**Figure 5 cancers-12-00929-f005:**
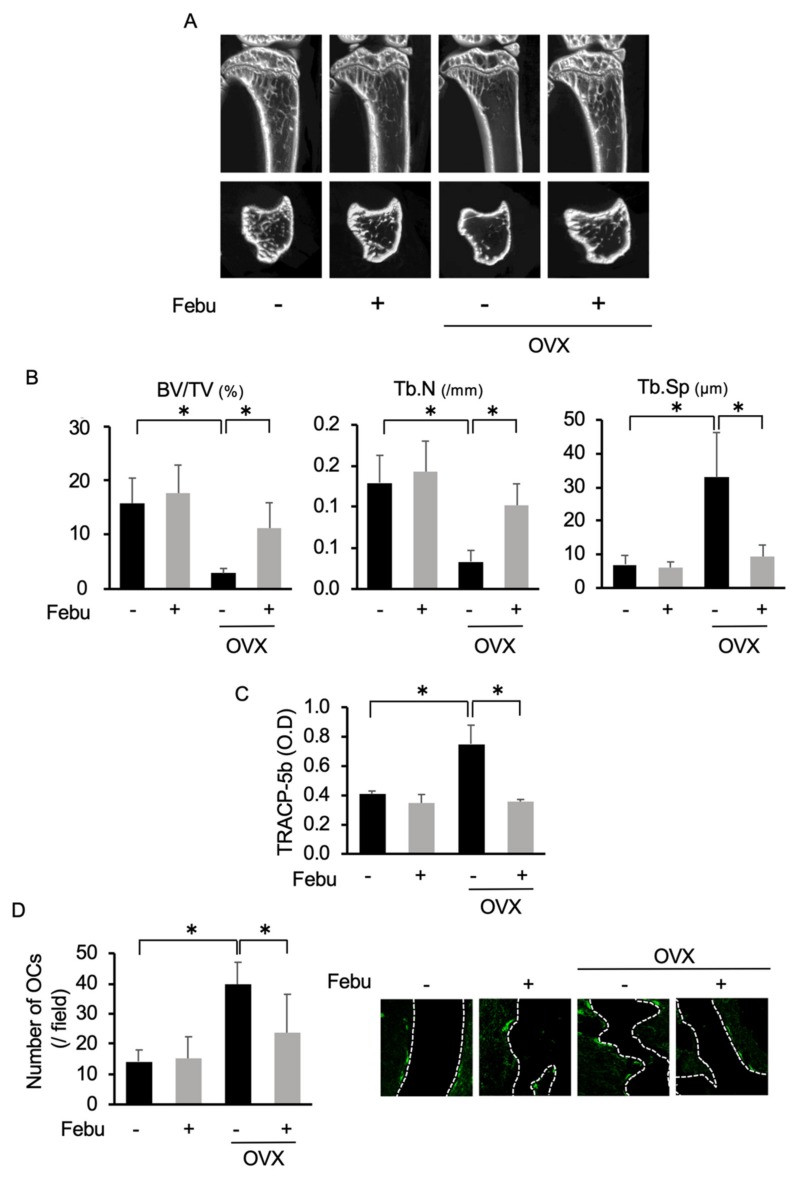
Febuxostat ameliorates bone loss in ovariectomized (OVX) mice. Sham-operated or OVX mice were treated for 3 weeks with or without febuxostat (Febu). μCT images were then taken. (**A**) Representative μCT images of sham and OVX mice with or without febuxostat (Febu) treatment are shown. (**B**) Bone volume/total volume (BV/TV), trabecular number (Tb.N), and trabecular separation (Tb. Sp) were calculated based on μCT images of the tibiae (*n* = 4 for each group). Results are expressed as mean ± SD. * *p* < 0.05. (**C**) The sera of sham and OVX mice were collected and serum levels of TRACP-5b were measured. Data are expressed as mean ± SD (n = 4 for each group). * *p* < 0.05. (**D**) The tissue sections of tibiae (*n* = 4 for each group) were stained with anti-cathepsin-K antibody to identify OCs, and OC numbers per field were counted. Data are expressed as mean ± SD. * *p* < 0.05. Representative photos are shown. Original magnification, ×100.

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
