# Peer review of "The Roles of ROS Generation in RANKL-Induced Osteoclastogenesis: Suppressive Effects of Febuxostat"

_cancers, 2020, doi:10.3390/cancers12040929_

Round 1
Reviewer 1 Report
In this manuscript, Ashtar et al. report the roles of ROS generation in RANKL-induced osteoclastogenesis in multiple myeloma. This paper is well written and will provide several important insights into understanding of bone lytic disease in patients with multiple meloma.
Specific points
Information about concentration of febuxostat in human patients will be useful. Figure 2C. 60 microM of febuxostat induced NFTAC1 expression in the absence of doxorubicin (DOX) and RANKL. The authors should explain the mechanism of this phenomenon. Figure 3B. Number of OCs was not affected by addition of DOX in the presence of RANKL (50 ng/mL). These results are not consistent with the results in Figure 2E. Please explain this difference. The effect of febuxostat on apoptosis of myeloma cells indeuced by DOX should be examined. Bortezomib is known to induce ROS in myeloma cells and it is interesting to know the effect of febuxostat on anti-myeloma effect of bortezomib.Author Response
Response to Reviewer1:
We thank the reviewer for raising important issues and the constructive comments. The comments were very helpful for improving our manuscript. A point-by-point response to the reviewer’s comments is listed below.
Information about concentration of febuxostat in human patients will be useful.
(Our response)
According to the drug information of febuxostat, Cmax and T1/2 are shown to be 1088.3 ng/ml (1088.3 ng/ml or 3440.0 microM) and 6.2 hour on average in normal subjects (n=8), respectively, after oral administration of febuxostat at 20mg.
Figure 2C. 60 microM of febuxostat induced NFTAC1 expression in the absence of doxorubicin (DOX) and RANKL. The authors should explain the mechanism of this phenomenon.
(Our response)
As the reviewer correctly pointed out, NFATc1 protein levels were increased upon treatment with febuxostat as well as NAC alone. However, osteoclastogenesis was not induced. We repeated the experiments but obtained the same results. However, NFATC1 mRNA expression levels were rather suppressed with febuxostat (Figure S1 in the revised manuscript). Because febuxostat and NAC are anti-oxidant agents, redox status may affect stabilization of NFATc1 protein. We need to further investigate the roles of febuxostat and NAC in nuclear localization and proteosomal degradation as well as expression by autoamplification in the presence or absence of DOX and/or RANKL. Because NFATc1 is one of the most critical factors in osteocalstogenesis, we plan to further study the expression and degradation of NFATc1 in osteoclastic cells in the presence of febuxostat or NAC. We added the description (lines 143-147 in the revised manuscript) as follows:
“Intriguingly, febuxostat as well as NAC induced NFATc1 expression in the absence of Dox and RANKL. However, NFATC1 mRNA expression levels were rather suppressed with febuxostat (Figure S1). Redox status under febuxostat or NAC may affect stabilization of NFATc1 protein, which should be further studied.”
Figure 3B. Number of OCs was not affected by addition of DOX in the presence of RANKL (50 ng/mL). These results are not consistent with the results in Figure 2E. Please explain this difference.
(Our response)
We studied the effects of Dox on osteoclastic differentiation in the presence of RANKL (Figure 2E, left) and those on viability of maturated osteoclasts already prepared on plastic dishes (Figure 3B in the original manuscript). We wanted to demonstrate that Dox does not impair the viability of osteoclasts while enhancing the differentiation into mature osteoclasts by RANKL.
The effect of febuxostat on apoptosis of myeloma cells induced by DOX should be examined. Bortezomib is known to induce ROS in myeloma cells and it is interesting to know the effect of febuxostat on anti-myeloma effect of bortezomib.
(Our response)
To examine the effect of febuxostat on apoptosis of myeloma cells, we looked at the activation of caspase3 in myeloma cells. Febuxostat did not compromise the activation of caspase3 induced by DOX. We included the results in Figure 3B in the revised manuscript, and the description (lines 187-188 in the revised manuscript) as follows:
“In addition, Dox induced the activation of caspase3 in MM cells, which was not compromised by febuxostat (Figure 3B).”
We also examined the effect of febuxostat on anti-MM effect of bortezomib and ixazomib. Febuxostat did not compromise anti-MM effects of bortezomib or ixazomib, while the ROS scavenger NAC compromised them as shown in Figure for Reviewer 1. We are interested in the difference between the xanthine oxidase inhibitors febuxostat or allopurinol and the ROS scavenger NAC especially in terms of effects on cytotoxic activity of anti-cancer agents, including proteasome inhibitors. We will further look into the underlying mechanisms of the difference between febuxostat and NAC in the next project.
Reviewer 2 Report
Comments to the authors
In this manuscript, the authors focused on ROS which is produced by anti-cancer drug and/or RANKL treatment, and addressed the effect of ROS inhibition on osteoclastogenesis by using febuxostat, a clinically used inhibitor of xanthine oxidase. The authors showed that febuxostat inhibited RANKL-induced and/or an anti-cancer drug Doxorubicin-induced production of ROS, and osteoclastogenesis along with induction of osteoclast differentiation markers. They also showed febuxostat treatment did not compromise the cytotoxic effect of Doxorubicin. Additionally, the authors showed the effect of febuxostat on bone marrow cell-MM coculture-induced osteoclastogenesis; febuxostat inhibited formation of osteoclasts.
As being submitted to Cancers, I think the contents should be focused on cancers; in this case, MM. However, the contents of this manuscript seem to be out of the focus, and I saw no mechanistic explanation. It is difficult to get massages from this manuscript, but I’d like to try criticize couple of points in this manuscript.
-Figure 1: Contents order seems messing up. A, B, D and G can be combined (or sequentially ordered). C seems having no mean here. Instead, it should be better to show picture images of osteoclast differentiation and resorption in F. Experimental time courses were different between RAW cells and BMMs, so it should be better show viability of BMMs after treatment with febuxostat.
-Figure 2: C and D showed that febuxostat and/or NAC treatment enhanced induction of NFATc1. This should be explained.
Regarding E, increased resorption area is obvious since the number of differentiated osteoclasts increased. It should be better to add additional data, that is, harvest differentiated osteoclasts and re-seed on to osteo-assay, so that resorption area per cell can be compared. Additionally, picture images are required for E.
-Figure 1 and 2: time courses are totally different between RAW cells and BMM-induced osteoclasts. Especially, it is unclear how Doxorubicin and/or febuxostat work on cells, because it took more than 10 days for induction of OC from BMMs. What about the effect of early phase treatment and/or late phase treatment?
-Figure 3: To emphasize the effect of febuxostat (it does not compromise the cytotoxic effects of doxorubicin), A should be better to become a single figure, and be separated from B and C. With regard to the sentence in discussion (line 284-285), it should be shown the effect of NAC, too.
Dose the Doxorubicin treatment affect ROS production from MM?
Regarding B, picture images are required.
Regarding C, I do not see importance of this data in the context of this manuscript. It can be move to supplementary data.
-Figure 4: this figure is confusing. In A, in the absence of MM cell line, some number of osteoclasts are generated. How could bone marrow cells become TRAP+ multinuclear cells without RANKL or VitD3 (if osteoblasts are source of RANKL)? How do TRAP+ multinuclear cells look in this experimental system? This should be explained, and picture images are also required.
In B, were “bone marrow stromal cells” derived from mouse? or human? (according to the method section, this looks like human cells, but not clear).
What about the effect of Doxorubicin on expression of RANKL on stromal cells?
GAPDH levels are different among samples, it should be adjusted. Or, Q-PCR or semi-quantitive PCR are required.
-Figure 5: Given that Doxorubicin has been used for treatment of multiple myeloma, it should be better to test the effect of febuxostat on mouse model of MM with treatment of Doxorubicin.
Regarding D, picture images are required.
-Abstract: contents order is totally different from the main text. It should be re-ordered.
Author Response
Response to Reviewer2:
We thank the reviewer for making important comments and valuable suggestions. The comments were very helpful for improving our manuscript. A point-by-point reply to the reviewer’s comments is listed below.
In this manuscript, the authors focused on ROS which is produced by anti-cancer drug and/or RANKL treatment, and addressed the effect of ROS inhibition on osteoclastogenesis by using febuxostat, a clinically used inhibitor of xanthine oxidase. The authors showed that febuxostat inhibited RANKL-induced and/or an anti-cancer drug Doxorubicin-induced production of ROS, and osteoclastogenesis along with induction of osteoclast differentiation markers. They also showed febuxostat treatment did not compromise the cytotoxic effect of Doxorubicin. Additionally, the authors showed the effect of febuxostat on bone marrow cell-MM coculture-induced osteoclastogenesis; febuxostat inhibited formation of osteoclasts.
As being submitted to Cancers, I think the contents should be focused on cancers; in this case, MM. However, the contents of this manuscript seem to be out of the focus, and I saw no mechanistic explanation. It is difficult to get massages from this manuscript, but I’d like to try criticize couple of points in this manuscript.
(Our response)
We aimed to focus on the cancer-associated and anti-cancer treatment-associated bone pathology in this manuscript, which is important for management of patients with bone cancers, including multiple myeloma (MM). RANKL-induced osteoclastogenesis is among predominant mechanisms of bone loss in MM. In this paper, we demonstrated that ROS is associated with RANKL-induced osteoclastogenesis, and suggest that anti-cancer agents may enhance RANKL-induced osteoclastogenesis. As the reviewer criticized, the mechanisms by which ROS facilitates RANKL signaling in osteoclastic cells and the suppression by febuxostat of induction of RANKL expression in bone marrow stromal cells largely remain to be clarified. We plan to dissect the underlying mechanisms in the next projects.
-Figure 1: Contents order seems messing up. A, B, D and G can be combined (or sequentially ordered). C seems having no mean here. Instead, it should be better to show picture images of osteoclast differentiation and resorption in F.
(Our response)
As the reviewer indicated, we deleted Figure 1C and its description (lines 95-98 in the original manuscript) and sequentially ordered A, B, D and G in the revised manuscript. Accordingly, we changed Figures 1D and 1G in the original manuscript to Figures 1C and 1D in the revised manuscript, respectively. We also included picture images of osteoclast differentiation and resorption in Figure 1F in the revised manuscript.
Experimental time courses were different between RAW cells and BMMs, so it should be better show viability of BMMs after treatment with febuxostat.
(Our response)
We added the results of the viability of BMMs after treatment with febuxostat in Figure 1D in the revised manuscript.
-Figure 2: C and D showed that febuxostat and/or NAC treatment enhanced induction of NFATc1. This should be explained.
(Our response)
As the reviewer correctly pointed out, NFATc1 protein levels were increased upon treatment with febuxostat as well as NAC alone. However, osteoclastogenesis was not induced. We repeated the experiments but obtained the same results. However, NFATC1 mRNA expression levels were rather suppressed with febuxostat (Figure S1 in the revised manuscript). Because febuxostat and NAC are anti-oxidant agents, redox status may affect stabilization of NFATc1 protein. We need to further investigate the roles of febuxostat and NAC in nuclear localization and proteosomal degradation as well as expression by autoamplification in the presence or absence of DOX and/or RANKL. Because NFATc1 is one of the most critical factors in osteocalstogenesis, we plan to further study the expression and degradation of NFATc1 in osteoclastic cells in the presence of febuxostat or NAC. We added the description (lines 143-147 in the revised manuscript) as follows:
“Intriguingly, febuxostat as well as NAC induced NFATc1 expression in the absence of Dox and RANKL. However, NFATC1 mRNA expression levels were rather suppressed with febuxostat (Figure S1). Redox status under febuxostat or NAC may affect stabilization of NFATc1 protein, which should be further studied.”
Regarding E, increased resorption area is obvious since the number of differentiated osteoclasts increased. It should be better to add additional data, that is, harvest differentiated osteoclasts and re-seed on to osteo-assay, so that resorption area per cell can be compared. Additionally, picture images are required for E.
(Our response)
Osteoclasts tightly stick onto plastic dishes, and are hard to come off from the dishes upon treatment with trypsin. We tried to collect osteoclasts on dishes prepared from BMMs in the presence of M-CSF and RANKL at day 7 with scraping after trypsin treatment, and then equally reseeded onto Osteo-Assay plates. The reseeded osteoclasts were cultured for 2 days in the presence of M-CSF, and their bone-resorbing activity was analyzed under treatment with RANKL and/or Dox in the presence or absence of febuxostat. Febuxostat clearly inhibited bone-resorbing activity of the reseeded osteoclasts, which was enhanced by RANKL or RANKL and Dox. However, addition of Dox did not further enhance bone resorptive activity in the presence of RANKL. Therefore, as the reviewer correctly criticized, the enhancement of bone resorptive activity by Dox (Figure 2E, right) appears to be due to that of osteoclast differentiation as shown in Figure 2E (left). We added the results in Figure S2 in the revised manuscript, and the description (lines 149-152 in the revised manuscript) as follows:
“However, addition of Dox did not enhance bone resorptive activity of re-plating osteoclasts at per cell levels in the presence of RANKL, while febuxostat was able to suppress bone resorbing activity of osteoclasts (Figure S2). Therefore, the enhancement of bone resorptive activity by Dox (Figure 2E, right) appears to be due to an increase in numbers of differentiated osteoclasts.”
-Figure 1 and 2: time courses are totally different between RAW cells and BMM-induced osteoclasts. Especially, it is unclear how Doxorubicin and/or febuxostat work on cells, because it took more than 10 days for induction of OC from BMMs. What about the effect of early phase treatment and/or late phase treatment?
(Our response)
RAW264.7 is a preosteoclastic cell line, which gives rise to a mature osteoclast faster than BMMs. In response to the reviewer’s criticism, we treated BMMs with doxorubicin and/or febuxostat for the first 2 days (days 1 and 2) or the later period, days 5 to 10. Treatment with febuxostat for each of these 2 limited periods was able to suppress osteoclast formation by RANKL. Treatment with doxorubicin from day 5 to day 10 enhanced osteoclast formation by RANKL, whereas the treatment for the first 2 days did not affect it. Precise mechanisms how doxorubicin work on osteoclastic cells remain to be clarified. We plan to study the time-course-dependent effects by doxorubicin as well as proteasome inhibitors on RANKL-mediated osteoclastogenesis in the next project. We added the results in Figure S3 in the revised manuscript, and the description (lines 153-158 in the revised manuscript) as follows:
“In addition, Treatment with febuxostat either for days 1 and 2 or for days 5 to 10 was able to suppress osteoclast formation by RANKL alone (Figure S3A). Treatment with doxorubicin from day 5 to day 10 enhanced osteoclast formation by RANKL, whereas the treatment for the first 2 days did not affect it (Figure S3B). Febuxostat also suppressed the Dox’s enhancement of osteoclast formation. Precise mechanisms how doxorubicin work on osteoclastic cells remain to be clarified.”
-Figure 3: To emphasize the effect of febuxostat (it does not compromise the cytotoxic effects of doxorubicin), A should be better to become a single figure, and be separated from B and C. With regard to the sentence in discussion (line 284-285), it should be shown the effect of NAC, too.
(Our response)
Thank you for your suggestion to emphasize the effect of febuxostat. We moved Figure 3B in the original manuscript to Supplementary Figure 5 in the revised manuscript. As the reviewer suggested, we examined the effects of NAC. Although febuxostat dose-dependently induced MM cell death and did not impair the cytotoxic effects of doxorubicin, NAC alone did not affect the viability of MM cells and appeared to compromise the cytotoxic effects of doxorubicin at the dose of 2.5 mM. We included the results with NAC as Figure 3C in the revised manuscript and added the description (lines 199-202) as follows:
“Although febuxostat dose-dependently induced MM cell death and did not impair the cytotoxic effects of doxorubicin, the ROS scavenger NAC alone did not affect the viability of MM cells and appeared to compromise the cytotoxic effects of doxorubicin at the dose of 2.5 mM (Figure 3C). These effects of febuxostat seems unique, which should be further studied.”
Dose the Doxorubicin treatment affect ROS production from MM?
(Our response)
Yes, doxorubicin treatment induced ROS production from MM cells, as shown in Figure S4. We added the description (line 181 ) in the revised manuscript as follows:
“Treatment with Dox induced ROS production from MM cells (Figure S4).”
Regarding B, picture images are required.
(Our response)
We added the photos of F-actin ring formed osteoclasts in the presence or absence of doxorubicin in Figure S5, because F-actin ring formation is a hallmark of intact mature osteoclasts.
Regarding C, I do not see importance of this data in the context of this manuscript. It can be move to supplementary data.
(Our response)
As the reviewer suggested, we moved Figure 3C in the original manuscript to Figure S6 in the revised manuscript.
-Figure 4: this figure is confusing. In A, in the absence of MM cell line, some number of osteoclasts are generated. How could bone marrow cells become TRAP+ multinuclear cells without RANKL or VitD3 (if osteoblasts are source of RANKL)? How do TRAP+ multinuclear cells look in this experimental system? This should be explained, and picture images are also required.
(Our response)
We used mouse whole bone marrow cells which contain RANKL-expressing bone marrow stromal cells. We added M-CSF for the first 3 days to prime osteoclastic differentiation of monocytes. After that, the cells were cultured alone or cocultured with MM cells in the presence of M-CSF. Fowler et al. reported similar experimental conditions to show osteoclast formation from mouse whole bone marrow cells (Journal of Cell Science (2015) 128, 683–694). We added the photos of TRAP+ multinuclear cells, and indicated the addition of M-CSF in the legend to Figure 4A.
In B, were “bone marrow stromal cells” derived from mouse? or human? (according to the method section, this looks like human cells, but not clear).
(Our response)
Thank you for your comment. Yes, they were derived from human. We indicated it in the legend to Figure 4B.
What about the effect of Doxorubicin on expression of RANKL on stromal cells?
(Our response)
We looked at the effects of Dox on RANKL mRNA expression in bone marrow stromal cells. Doxorubicin at 0.5 μM did not induce the expression of RANKL mRNA in bone marrow stromal cells. We added the result as Figure 4C and the description (lines 228-229) in the revised manuscript as follows:
“Doxorubicin alone did not induce the expression of RANKL mRNA in bone marrow stromal cells.”
GAPDH levels are different among samples, it should be adjusted. Or, Q-PCR or semi-quantitative PCR are required.
(Our response)
We reexamined the experiment and replaced the figure.
-Figure 5: Given that Doxorubicin has been used for treatment of multiple myeloma, it should be better to test the effect of febuxostat on mouse model of MM with treatment of Doxorubicin.
(Our response)
Yes, it should be better to test the effect of febuxostat on mouse model of MM with treatment of doxorubicin. We plan to examine the effect of febuxostat or NAC on bone lesions and tumor progression with or without combination with proteasome inhibitors as well as doxorubicin in the next project.
Regarding D, picture images are required.
(Our response)
We included the immunofluorescent staining with anti-cathepsin K antibody in Figure 5D in the revised manuscript.
-Abstract: contents order is totally different from the main text. It should be re-ordered.
(Our response)
We re-ordered Abstract as follows:
“Abstract: Receptor activator of NF-κB ligand (RANKL), a critical mediator of osteoclastogenesis, is upregulated in multiple myeloma (MM). The xanthine oxidase inhibitor febuxostat, clinically used for prevention of tumor lysis syndrome, has been demonstrated to effectively inhibit not only the generation of uric acid but also the formation of ROS. Reactive oxygen species (ROS) has been demonstrated to mediate RANKL-mediated osteoclastogenesis. In the present study, we therefore explored the role of cancer treatment-induced ROS in RANKL-mediated osteoclastogenesis and the suppressive effects of febuxostat on ROS generation and osteoclastogenesis. RANKL dose-dependently induced ROS production in RAW264.7 preosteoclastic cells; however, febuxostat inhibited the RANKL‐induced ROS production and OC formation. Interestingly, doxorubicin (Dox) further enhanced RANKL-induced osteoclastogenesis through upregulation of ROS production, which was mostly abolished by addition of febuxostat. Febuxostat also inhibited osteoclastogenesis enhanced in cocultures of bone marrow cells with MM cells. Febuxostat rather suppressed MM cell viability and did not compromise Dox’s anti-MM activity. In addition, febuxostat was able to alleviate pathological osteoclastic activity and bone loss in ovariectomized mice. Collectively, these results demonstrate that excessive ROS production by aberrant RANKL overexpression in MM and/or anti-cancer treatment disadvantageously impacts bone, and that febuxostat can prevent the ROS-mediated osteoclastic bone damage.”
Reviewer 3 Report
In the present study, the authors evaluated the effects of febuxostat on osteoclastogenesis; in particular, they demonstrated that Febuxostat suppressed the ROS production and osteoclastogenesis by Dox and RANKL in combination. Febuxostat counteracted osteoclastogenesis enhanced in cocultures of bone marrow cells with myeloma cells; finally, they performed in vivo studies demonstrating a reduction of bone loss in ovariectomized-mice, after Febuxostat treatment.
Overall, manuscript is well written and the experimental data are well investigated and exhaustive. In my opinion, the manuscript can be considered acceptable after minor revision.
Co-culture methods are not reported in the section ‘Material and Methods’ (effects of febuxostat on in vitro osteoclastogenesis in cocultures of MM cells with primary bone marrow cells.). Authors need to clearly explain as they performed co-culture experiments. In the paragraph of Results about ‘ Febuxostat suppresses osteoclastogenesis by MM cells’, the authors report that ‘MM cells upregulate RANKL expression in the bone marrow stroma cells to enhance osteoclastogenesis’. Nonetheless, in the reference indicated by the authors [29: Kondo Y et al., J. Bone Miner. Metab. 2001] this experimental result is not reported. Authors must carefully review the references indicated. The authors report in Figure legend 2E: ‘The numbers of TRAP-positive multinucleated cells with 6 or more nuclei were counted per well’. Why do the authors consider differentiated cells with 6 or more nuclei? Is a mistake? Actin ring formation is a prerequisite for osteoclast bone resorption; mature osteoclasts display a full actin ring or a disrupted actin rings when differentiation is pertubated. The images observed by fluorescence microscope (cells fixed and stained with Acti-stain488 phalloidin) should be showed by the authors, in addition to the data shown in the Figure 3B (right). Authors report data on PIM2 expression under Dox treatment; they found that Dox dose-dependently induced the expression of PIM2 in mature OCs as RANKL (Figure 3C). According to me, results about PIM2 expression are not particularly relevant for this study; authors should move these results into the Supplementary section.Author Response
Response to Reviewer3:
We thank the reviewer for making critical comments and valuable suggestions. The comments were very helpful for improving our manuscript. A point-by-point reply to the reviewer’s comments is listed below.
Co-culture methods are not reported in the section ‘Material and Methods’ (effects of febuxostat on in vitro osteoclastogenesis in cocultures of MM cells with primary bone marrow cells.). Authors need to clearly explain as they performed co-culture experiments.
(Our response)
We are sorry, but we added the description (lines 356-362) in the section ‘Material and Methods’ in the revised manuscript as follows: “Whole bone marrow cells harvested from C57BL/6J mice were seeded onto 24-well plate and cultured with M-CSF (10 ng/mL) to generate osteoclast precursors for 3 days. And then, human MM cell line, RPMI8226 and KMS-11, murine MM cell line, 5TGM1 were added at 1×104 cells per well. The cells were cultured with or without febuxostat at 60 μM for 14 days in the presence of M-CSF (10 ng/mL). The numbers of TRAP-positive multinucleated cells per well were counted under a light microscope (BX50; Olympus, Tokyo, Japan).”
In the paragraph of Results about ‘ Febuxostat suppresses osteoclastogenesis by MM cells’, the authors report that ‘MM cells upregulate RANKL expression in the bone marrow stroma cells to enhance osteoclastogenesis’. Nonetheless, in the reference indicated by the authors [29: Kondo Y et al., J. Bone Miner. Metab. 2001] this experimental result is not reported. Authors must carefully review the references indicated.
(Our response)
We are sorry for our mistake. As the reviewer correctly criticized, we replaced the reference #29 with the reference below:
Giuliani N, Colla S, Rizzoli V. New insight in the mechanism of osteoclast activation and formation in multiple myeloma: focus on the receptor activator of NF-κB ligand (RANKL). Exp. Hematol. 2004;32: 685-691.
The authors report in Figure legend 2E: ‘The numbers of TRAP-positive multinucleated cells with 6 or more nuclei were counted per well’. Why do the authors consider differentiated cells with 6 or more nuclei? Is a mistake?
(Our response)
Osteoclasts fuse with each other to have more nuclei (6 or more), as they mature. When mature osteoclasts with a larger number of nuclei increase, increase in total TRAP-positive multinucleated osteoclast numbers becomes rather blunted. Therefore, we counted the numbers of multinucleated cells with 6 and more nuclei to better demonstrate the effects on maturation of osteoclasts (Figure 2E, left). We added the numbers of TRAP-positive multinucleated osteoclasts with 3–5 nuclei in Figure 2E (left) in the revised manuscript.
Actin ring formation is a prerequisite for osteoclast bone resorption; mature osteoclasts display a full actin ring or a disrupted actin rings when differentiation is perturbated. The images observed by fluorescence microscope (cells fixed and stained with Acti-stain488 phalloidin) should be showed by the authors, in addition to the data shown in the Figure 3B (right).
(Our response)
We added the images of actin ring formation stained with Acti-stain488 phalloidin by a fluorescence microscope to Figure S5 in the revised manuscript. In response to the suggestion by Reviewer2, we moved Figure 3B in the original manuscript to Figure S5 in the revised manuscript.
Authors report data on PIM2 expression under Dox treatment; they found that Dox dose-dependently induced the expression of PIM2 in mature OCs as RANKL (Figure 3C). According to me, results about PIM2 expression are not particularly relevant for this study; authors should move these results into the Supplementary section.
(Our response)
As the reviewer kindly suggested, we moved Figure 3C to Figure S6.
Reviewer 4 Report
This study investigated therapeutic utility of the xanthine oxidase (XO) inhibitor, febuxostat on reactive oxygen species (ROS) production during RANKL induction of osteoclast (OCL) differentiation. Previous studies by Kurihara and colleagues have shown that blocking sequestosome suppresses OCL formation and induce bone formation in myeloma tumor bearing mice. The manuscript is well written and conclusive of results that febuxostat prevent the ROS mediated OCL bone destruction and appears to be an effective therapeutic agent against myeloma bone disease.
This study is unique that Febuxostat, a selective and potent inhibitor of XO suppress OCL formation. Since ROS is produced from different sources such as NADPH oxidase, xanthine oxidase, and mitochondrial electron transport system (Shin et al., Free Radical Biol and Med, 44:635, 2008), they may discuss the implications of the study to use an agent which can suppress ROS produced from all sources other than xanthine oxidase against myeloma.
Please note the statistical significance (*) in Fig.1A-F graphs like in other figs.2,4, and 5 shown.
(p.10; lines 264-265)- They have noted “…..ROS generation in osteoclastic lineage cells by Dox…..” Please verify/remove Dox, since it has shown to have anti-osteoclastogenic activity (Naghsh et al., 13:500, 2016).
Author Response
Response to Reviewer4:
We thank the reviewer for making thoughtful comments and valuable suggestions. The comments were very helpful for improving our manuscript. A point-by-point reply to the reviewer’s comments is listed below.
Since ROS is produced from different sources such as NADPH oxidase, xanthine oxidase, and mitochondrial electron transport system (Shin et al., Free Radical Biol and Med, 44:635, 2008), they may discuss the implications of the study to use an agent which can suppress ROS produced from all sources other than xanthine oxidase against myeloma.
(Our response)
Because febuxostat is now clinically used for prevention of tumor lysis syndrome in patients with malignant tumors receiving chemotherapy, we focused on febuxostat in the present study. However, as the reviewer indicated, ROS is produced from different sources such as NADPH oxidase and mitochondrial electron transport system other than xanthine oxidase.
Therefore, we need to study the mechanisms of ROS production in MM cells and osteoclasts under different stimuli, and the effects of agents to suppress ROS produced from all sources other than xanthine oxidase. We added the description in Discussion (lines 300-303 in the revised manuscript) as follows:
“In addition, ROS is produced from different sources such as NADPH oxidase and mitochondrial electron transport system other than xanthine oxidase. Therefore, we need to clarify the mechanisms of ROS production in MM cells and osteoclasts under different stimuli, and also the effects of agents to suppress different sources of ROS production other than xanthine oxidase.”
Please note the statistical significance (*) in Fig.1A-F graphs like in other figs.2,4, and 5 shown.
(Our response)
We added the statistical significance (*) in Fig.1 in the revised manuscript.
(p.10; lines 264-265)- They have noted “…..ROS generation in osteoclastic lineage cells by Dox…..” Please verify/remove Dox, since it has shown to have anti-osteoclastogenic activity (Naghsh et al., 13:500, 2016).
(Our response)
As the reviewer kindly suggested, we remove Dox:
“However, febuxostat was shown to effectively inhibit ROS generation in osteoclastic lineage cells by Dox as well as RANKL, and suppress the induction of osteoclastogenesis by Dox and RANKL in combination.” was changed to “However, febuxostat was shown to effectively inhibit ROS generation in osteoclastic lineage cells by RANKL, and suppress the induction of osteoclastogenesis by RANKL.” in the revised manuscript.
Round 2
Reviewer 2 Report
The authors addressed and answered questions by performing additional experiments. In addition, although the mechanism underlying the effect of febuxostat is still largely unclear, they added discussion about this issue in main text. I’m satisfied the revised manuscript.